# Novel brain biomarkers of obesity in young adult women based on statistical measurements of white matter tracts

**José Gerardo Suárez-García**[1], **María Isabel Antonio-de la Rosa**[1],
**Nora Coral Soriano-Becerril**[2], **Javier M. Hernández López**[1],
**Martín Rodolfo Palomino-Merino**[1], **Benito de Celis-Alonso**[1]*

**1** Faculty of Physical and Mathematical Sciences, Benemérita Universidad Autónoma de Puebla (BUAP), Puebla, Mexico, **2** Instituto Mexicano del Seguro Social (IMSS), Puebla, Mexico

* bdca@fcfm.buap.mx

## Abstract

### Objective

Novel brain biomarkers of obesity were sought by studying statistical measurements on fractional anisotropy (FA) images of different white matter (WM) tracts from young adult women.

### Methods

Tract measurements were chosen that showed differences between two groups (normal weight and overweight/obese) and that were correlated with BMI. From these measurements, a simple and novel process was applied to select those that would allow the creation of models to quantify and classify the state of obesity of individuals. The biomarkers were created from the tract measurements used in the models.

### Results

Positive correlations were found between WM integrity and BMI, mainly in tracts involved in motor functions. From these results, two models were built to quantify and classify obesity status, whose regression coefficients formed the novel proposed obesity associated brain biomarkers.

### Conclusion

A process for the selection of tract measurements was proposed, such models were built to determine the obesity status of subjects individually. From these models, novel brain biomarkers associated with obesity were created. These results generate new knowledge in the field, intended to be used in the future in the clinical environment as a prevention and treatment tool for brain changes associated with obesity.

**Data availability statement:** All relevant data are within the paper and its Supporting Information files.

**Funding:** The author JGSG was supported by the National Council of Humanities, Sciences, and Technologies (CONAHCYT) (https://conahcyt.mx), to carry out this work, through a posdoctoral scholarship. The funders had no role in study design, data collection and analysis, decision to publish, or preparation of the manuscript.

**Competing interests:** The authors have declared that no competing interests exist.

## Significance

After studying young adult women, results opposed some of the previous results reported in literature. These consisted of positive correlations between WM integrity and obesity mainly in tracts involved in motor functions. Novel brain biomarkers of obesity were also proposed, formed by the regression coefficients involved in precise models of quantification and classification of obesity status. All this allows the generation of new knowledge and its probable subsequent clinical application.

## 1. Introduction

Obesity is a major health problem worldwide. In 2022, 2.5 billion adults in the world (which corresponds to 43% of the total) were overweight, including over 890 million adults who were living with obesity [1]. It is known that obesity induces a myriad of pathological alterations that include between others: neuroinflammation, vascular damage, metabolic imbalances, blood–brain barrier disruption, as well as being a precursor to some types of cancers [2]. In the neurological context, several works have found associations between obesity and the structure and function of the gray (GW) and white matter (WM). This by analyzing multi-modal magnetic resonance imaging (MRI), positron emission tomography (PET), and single-photon emission computed tomography (SPECT) images [3]. Regarding WM, diffusion tensor imaging (DTI) is an MRI modality that allows the study of the integrity and coherence of WM tracts through a scalar measurement named fractional anisotropy (FA). It measures the water diffusivity along the axons that form the WM tracts, with high FA values representing highly organized and normally myelinated axon structures [4]. On the other hand, reduced FA values can be interpreted as a loss of coherence in the main preferred diffusion direction, resulting in a deficit in white matter microstructure and integrity. FA has become the most used DTI measurement to quantify WM characteristics, specifically in voxel-based analysis (VBA) and tract-based and spatial statistic (TBSS) [5].

Most works studying FA have found that adults with a higher BMI have poorer WM integrity, indicating that the fiber tracts connecting various brain regions are less efficient and slower during the neural transmission and information processing [6]. In a recent work, Dietze et al. [7] included 51 studies in a voxel based meta-analysis. They reported an association between reduced FA and obesity in the genu and splenium of the corpus callosum, middle cerebellar peduncles, anterior thalamic radiation, corticospinal projections and cerebellum. Also, their findings demonstrated that, obesity related brain white matter changes, were localized rather than diffuse. In another work, Chen et al. [8] conducted a coordinate-based meta-analysis on 5 studies. Their findings showed that overweight or obese individuals exhibited reduced FA compared to subjects with normal weight, in the right superior longitudinal fasciculus, the splenium of the corpus callosum, the right median network and cingulum. Besides, Okudzhava et al. [9] conducted a systematic review considering 31 studies, finding that the majority reported decreased FA associated with elevated BMI.

Despite all the evidence presented on decreased FA associated with increased BMI, other studies found, either no significant differences between FA values and obesity measurements, or positive associations between both in some tracts [5]. Verstynen et al. [10] found that the FA of WM voxels parametrically decreased with increasing BMI. However, several clusters with strong positive relationships were also reported in the middle of the cerebellar peduncle and the fiber pathways near the colliculus. Birdsill et al. [11] found that higher waist circumference was associated with higher FA in the posterior white matter bilaterally. Based on their

results, the authors speculated that possible degenerative processes could account for the positive association found, such as myelin repair, loss of long white matter tracts with sparing of short interneurons, and a lack of reorganization.

Among the works reported in the literature, it is common to study wide age ranges and volunteers of both sexes. Few works have studied men and women separately. Currently, there is a demand from all funding agencies to include sex as a biological factor and provide further support for the importance of including sex-based analyses, as females and males may exhibit differences in neural mechanisms [12,13]. Rahmania et al. [14] investigated whether gender in mid-life differentially affects the relationship between BMI and WM structural connectivity. In contrast to men, women (of all BMIs) showed a relationship between BMI and increased connectivity in several WM tracts, including the bilateral frontoparietal cingulum, bilateral reticulospinal, dentatorubrothalamic, corticopontine tracts, the left corticospinal tract and the tapetum area of corpus callosum. Here, authors discuss that higher tissue adiposity conferred a protective effect of the WM connectome in normal to overweight women due to the neuro-protective effects of estrogen on WM.

In addition to studying wide age ranges and mixed sexes, previous works have reported findings at the group level and not individually. This implies that those results provide statistical information about the groups studied, but this information does not always allow obtaining precise information for individual subjects. Therefore, biomarkers whose purpose was to associate specific characteristics with individual subjects, cannot be proposed based on this information obtained at the group level. However, in literature one can find works whose characterization on individuals has been attempted by creating models using linear regression, neural networks, etc., that combine different quantitative characteristics of the brain (structural or functional) to predict the obesity status of individual subjects [2,3,15,16]. The objective of these models was not to use a complex methodology to diagnose the obesity status of a subject, since this can be trivially measured. Their objective was to discover relationships between brain parameters measured, and the already known obesity status of an individual. Subsequently and based on those relationships, authors proposed obesity biomarkers.

Considering that for the proposal of biomarkers there are few studies considering sex and individual subject information vs. grouped data, in the present work the main objective was to search for novel obesity biomarkers, specifically associated with young women. The biomarkers were formed by a combination of statistical measurements made on WM tracts considering the voxel FA values. From here on these measurements will be named "tract measurements". To carry out this project, the protocol was divided into two parts to achieve two specific objectives. In the first part, the objective was to find tracts that were associated with obesity, paying special attention to those reported in the literature that showed some differences associated with women compared to men. Among these were: the corpus callosum, middle cerebellar peduncles, anterior thalamic radiation, corticopontine tracts, corticospinal tracts, and cingulum cingulate [12,14]. Despite this, all brain tracts were equally studied. To achieve this, different tract measurements were chosen when they met two conditions: First, to present significant differences between the two study groups (obese (OB) and normo-weight (NW)) (condition one), and second, to be significantly correlated with the subject's BMI (condition two). This could be considered as the extraction of information at group level (since it was obtained from comparing two study groups). The hypotheses associated to the first objective was that, compared to that reported usually by other works where they study a wide range of ages and mixed sexes, in this work different findings at the group level will be only young adult women. The findings will be based on tract measurements that, in this work, will be associated with obesity, studying all WM tracts, paying special attention to those that have been reported as specifically related to women, as well as others that have not been. In

the second part, the objective was constructing novel biomarkers based on the tracts measurements that were found to be associated with obesity in young adult women in the first part of the work. Using the chosen tract measurements, two regression models were built to predict the obesity status of the subjects: One quantifying the BMI and the other classifying subjects into the OB or NW groups. This could be considered as obtaining information at individual level (since the obesity status was predicted individually for each subject). Then, obesity biomarkers were constructed by combining the tract measurements used in the models. In this part of the work, it is hypothesized that the construction of novel biomarkers associated with obesity in young adult women, built from tract measurements involved in the predicting models, will be possible. In general, all work aims look to provide new knowledge on changes in WM associated with obesity, specifically in young adult women.

## 2. Methodology

In the first part of the work, tract measurements were obtained by calculating different statistics on the FA data of each WM tract. Then, among all the measurements, those that met two conditions were chosen. In the second part, considering the previously chosen tract measurements, two regression models were built. A novel methodology was proposed to select tract measurements that would allow the creation of accurate models. Finally, new obesity biomarkers were proposed. Unless otherwise stated, algorithms were developed in MATLAB R2023b, using a conventional computing system (Intel Core i7-12700H, NVIDIA GeForce RTX 3070 Ti, 32 GB RAM).

### 2.1. Database

The open-access database Amsterdam Open MRI Collection (AOMIC) was used for this study, which consisted of three large-scale datasets with high-quality, multimodal 3T MRI data and detailed demographic and psychometric data from a large set of healthy participants. Of the three datasets, the so-called "ID1000" was studied [17]. AOMIC contains both raw data as well as preprocessed data from well-established preprocessing and quality control pipelines. Among the different available MRI modalities, preprocessed diffusion-weighted MRI (DWI) were considered in the present work, from which data derived from the original raw data consisting of fractional anisotropy (FA) maps were studied. Basic information about the database will be summarized below, although a more detailed description can be found on the database website [18], and in a paper published in Nature Scientific Data [19].

Since this work studied subjects with different BMIs, including those categorized as overweight or obese, it should be clarified that the term "healthy," as referred to by the creators of the AOMIC database, means that the subjects did not report any known pathology or comorbidity associated with their respective overweight or obese status. On the other hand, in this work the authors recognize that a state of overweight or obesity by itself can be considered as an unhealthy state.

**2.1.1. MRI scanning protocol.** Based on the description included in the database website and in its respective publication [17,18], data from ID1000 dataset were scanned on a Philips 3T scanner (Philips, Best, the Netherlands), on the "Intera" version using a 32-channel head coil. At the start of each scan session, a low-resolution survey scan was made, which was used to determine the location of the field-of-view. Three $T_1$-weighted scans, three diffusion-weighted scans, and one functional (BOLD) MRI scan were recorded (in that order). For all diffusion scans, the slice stack was not angled. Three scans were obtained with the SE-DWI technique with a b0 image, 32 diffusion-weighted directions, a half sphere sampling scheme, and DWI b-value equal to 1000 s/mm$^2$. Voxel size was equal to $2 \times 2 \times 2$ mm, FOV

of $224 \times 224 \times 120$, matrix size of $112 \times 112$, 60 slices with no slice gap, TR = 6370 ms and TE = 75 ms, water-fat shift of 12,861 pixels, bandwidth equal to 33.8 Hz/pixel, flip angle of 90 degrees and with a duration of 4 minutes and 49 seconds.

**2.1.2. DWI standardization, preprocessing and FA image computing.** Preprocessing was already implemented on the database by creators. A summary of the processes applied are presented below, although a more detailed description of the entire preprocessing can be found on the database website and in its respective publication [17,18]. According to the authors of the database, data were converted to BIDS, including file renaming, conversion to compressed nifti, and defacing and extraction of metadata. The three DWI scans per participant, the diffusion gradient table, and b-value information were concatenated. Following this, preprocessing was applied to the data using tools from MRtrix3 and FSL. This consisted of denoising the diffusion-weighted data using *dwidenoise* [20,21], removing Gibbs ringing artifacts using *mrdegibbs* [22], and performing eddy current and motion corrections using *dwipreproc*. Within the eddy, a quadratic first-level and linear second-level model and outlier replacement with default parameters were used. Bias correction and brain mask extraction were also performed. To validate the consistency in the data preprocessing steps using MRtrix3 and FSL software, specifically regarding checking the orientation of the diffusion gradient table, the database authors used *dwigradcheck* to correct possible problems of improperly rotated diffusion gradient orientations in diffusion weighted MRI. This algorithm is based on the method proposed by Jeurissen et al. [23]. A diffusion tensor model on the preprocessed diffusion-weighted data using weighted linear least squares with 2 iterations was fit using *dwi2tensor* [24]. From the estimated tensor image, a fractional anisotropy (FA) image was computed and a map with the first eigenvectors was extracted using *tensor2metric*.

**2.1.3. Affine aligned into MNI152 standard space.** In the present work, an additional affine alignment of the FA images into MNI152 standard space was performed. To this end, two FSL scripts available online (developed originally to perform TBSS) were applied [25]. The first script was *tbss_2_reg*, used to align all FA images to a 1x1x1mm standard space by performing nonlinear registration and considering the adult-derived target image FMRIB58_ FA. The second script was *tbss_3_postreg*, which made nonlinear transformations to bring the images into MNI152 standard space.

**2.1.4. Subjects included in the study.** Data from 992 subjects (men and women) were available within the ID1000 dataset. According to the authors, subject sample was representative of the general Dutch population in terms of educational level (as defined by the Dutch government), and with age range from 19 to 26 years to minimize the effect of aging on any brain-related covariates. Subjects' body-mass-index (BMI) was calculated and rounded to the nearest integer. Educational level was reported on a three-point scale: low, medium, and high, based on the completed or current level of education. In the present work, to reduce the variability between the demographic characteristics of the subjects, only right-handed females with a medium level of education were considered (including upper secondary education (HAVO/VWO), basic vocational training (MBO-2), vocational training (MBO-3), and middle management and specialist education (MBO-4), based on the Dutch education system [26]) and with an BMI over $19 \, \text{kg/m}^2$ (excluding subjects with low weight). In total, 160 subjects met the criteria for inclusion (S1 and S2 Tables). The set formed by all the subjects was called $D_{all}$. From this, two subsets were created, one called $D_{norm}$, consisting of 80 subjects with $19 \, \text{kg/m}^2 \leq \text{IBM} < 25 \, \text{kg/m}^2$ (NW), and another called $D_{over}$, consisting of 80 subjects with $25 \, \text{kg/m}^2 \leq \text{IBM} \leq 47 \, \text{kg/m}^2$ (OB, with $47 \, \text{kg/m}^2$ being the highest IBM available). Information on $D_{norm}$ and $D_{over}$ subsets is shown in Table 1.

**Table 1. Information about $D_{norm}$ and $D_{over}$ subsets.** The number of subjects according to their weight classification in each subset, as well as average age and BMI are shown.

| | **Dnorm** | **Dover** | | |
|---|---|---|---|---|
| **Subjects** | Normal | Overweight | Obese | Extremely obese |
| | $19\,kg/m^2 \leq BMI < 25\,kg/m^2$ | $(25 \leq BMI < 30)$ | $(30 \leq BMI < 35)$ | $(35 \leq BMI \leq 47)$ |
| | 80 (total) | 49 | 19 | 12 |
| | | 80 (total) | | |
| **Age (years)** | $22.65 \pm 1.66$ | $22.63 \pm 1.71$ | | |
| **BMI (kg/m²)** | $21.63 \pm 1.72$ | $29.55 \pm 4.63$ | | |

## 2.2. Part 1: Choosing tract measurements

For the analysis of the tracts, the ICBM-DTI-81 white-matter labels atlas (S3 Table) was used, which is composed of 50 tracts. For each of the subject tracts, 12 measurements corresponding to descriptive statistics were calculated (S4 Table) considering the FA values of the voxels that formed them. Thus, a total of 600 tract measurements were obtained per subject. The type of analysis of the present work was, therefore, a VBA. Subsequently, for each tract measurement, the Wilcoxon rank sum test was applied to the tract measurements from the $D_{norm}$ and $D_{over}$ subsets to determine whether there were significant differences ($p_w < 0.05$) between both. If there was a significant difference, then the respective tract measurement met condition one. Also, for each tract measurement, the Spearman correlation coefficient ($\rho$) and its significance ($p_c$) were calculated between the tract measurements and the BMI of the subjects that formed $D_{all}$. If there was a significant correlation ($p_c < 0.05$, FDR-corrected), then the respective tract measurement met condition two. Tract measurements that met both conditions were chosen, and the total number was called $N$. In the second part of the work, these $N$ tract measurements were used to build the two regression models mentioned above.

## 2.3. Part 2: Regression models and creation of biomarkers

Before creating any model, the $D_{all}$ set was separated into three subsets called $D_{tr}$, $D_{val}$, and $D_{te}$, corresponding to training, validation and testing subsets, and formed by 80, 40, and 40 subjects respectively. Each subset included NW and OB subjects chosen randomly, but under the condition that the three subsets had approximately a homogeneous distribution of the different BMI values available. When possible, the training subset contained twice the number of subjects with the same BMI contained in the validation or testing subsets. The distribution of subjects by BMI in each subset is shown in S2 Fig.

**2.3.1. Model 1: Quantification of BMI.** Model 1 aimed to individually predict the obesity level or status of each subject by predicting a quantity for their BMI. The process for creating models by randomly adding and excluding tract measurements was intended to create progressively better models. In Fig 1 a diagram schematically representing the process is shown, where the blue shaded area represents the process of adding and the yellow shaded area represents the process of excluding measurements. A step was defined as each occasion on which a new model obtained the best results up to that moment. The proposed process will be described below. From the $N$ tract measurements, one was randomly chosen, and a simple linear regression model was created and adjusted considering the subset $D_{tr}$. The model was applied to the subjects of $D_{tr}$ and $D_{val}$ to predict their BMI. Pearson correlation coefficients $r_{train}$ and $r_{val}$ were calculated by comparing the actual and predicted BMI values. From $r_{train}$ and $r_{val}$, an average coefficient $r_{mean}$ was calculated. This model being the first one created, it was considered the best one up to that point with the coefficient of the highest value, thus having the first step. The change $r_{mean} > r_{mean}^*$, was made, where the asterisk indicated that

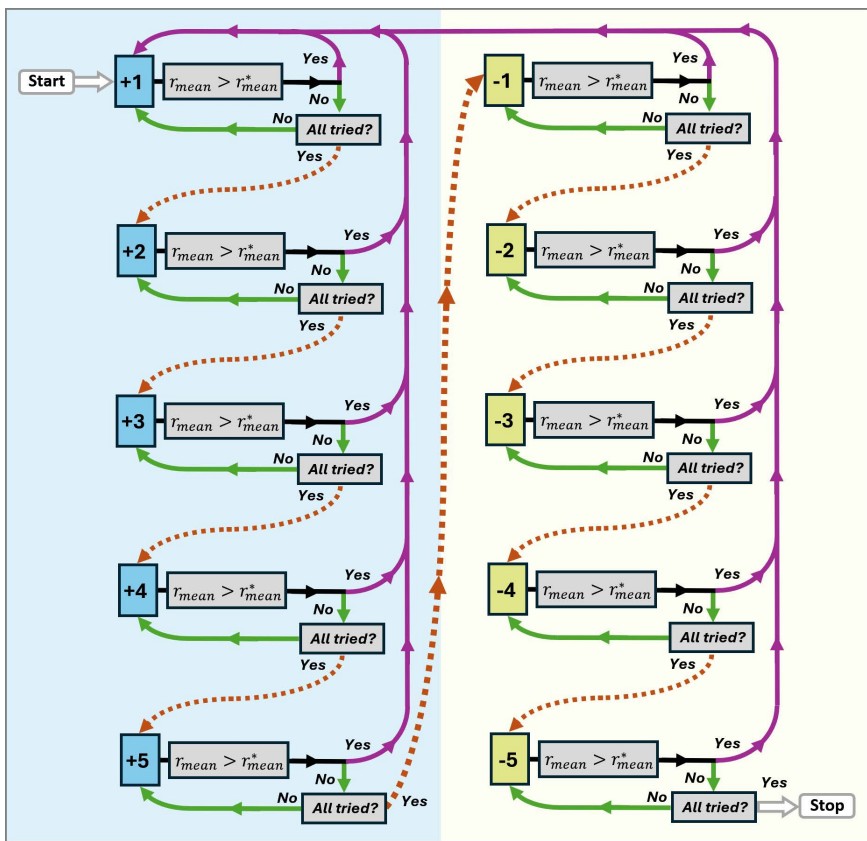

**Fig 1. Process for creating model 1 to quantify BMI.** This process consisted of progressively adding (+) or excluding (-) from 1 to 5 tract measurements at a time to create new and better models. An average correlation coefficient $r_{mean}$ was calculated and the condition $r_{mean} > r_{mean}^*$ was evaluated, with $r_{mean}^*$ being the highest coefficient at the time of evaluation. As long as the condition was met, the aggregation or exclusion was maintained, and the process was restarted (purple solid arrow, Start). Otherwise, aggregation or exclusion was not maintained and the search continued until the available tract measurements were exhausted (green solid arrow), considering from one to five measurements at a time (orange dotted arrow). The entire process ended when no better results were obtained (Stop).

this coefficient corresponded to the highest so far. From the remaining $N – 1$ measurements, another one was randomly chosen, and together with the first one, a new two-variable linear regression model was created using $D_{tr}$. Similarly to before, the new model was applied to the subjects from $D_{tr}$ and $D_{val}$, and the respective coefficients $r_{train}$, $r_{val}$ and $r_{mean}$ were calculated. The following condition was evaluated: $r_{mean} > r_{mean}^*$, where $r_{mean}$ was the coefficient of the last model created and $r_{mean}^*$ was the coefficient of the best model so far. If the condition was met, then the second step was obtained and the following update was made: $r_{mean}^* = r_{mean}$. If the condition was not met, the $r_{mean}^*$ coefficient was not updated, there was no new step, and the last measurement added was not considered for the following models to be built.

Subsequently, another measurement was randomly chosen, a new model was created with the previous measurements and the new one chosen was applied to the $D_{tr}$ and $D_{val}$ subjects, the respective average coefficient $r_{mean}$ was calculated and the condition $r_{mean} > r_{mean}^*$ was evaluated. This process of randomly searching for a new measurement and, together with the previous ones, creating a new model, was repeated until all the available measurements were considered without repeating any of them. Then, a process similar to the one described above was repeated by searching and adding from two to five measurements at a time. If adding a

number of measurements at a time yielded better results, the search was repeated by searching and adding only one measurement at a time.

If the search for five measurements at a time to be added to the best model was exhausted, a different process was carried out in which measurements were excluded from those already added. This consisted of the following. Assuming that the last best model consisted of $m$ measurements, then a measurement was randomly chosen from the $m$ already added. This measurement was excluded from the model, a new model with $m - 1$ measurements was created, the respective coefficients were calculated and the condition was evaluated in the same way as before. If a new model with better results was created, a new step was obtained and the entire process of searching and adding a single measurement at a time was repeated. If no better results were obtained, the process of excluding a measurement from those already added continued until better results were obtained. If all measurements were exhausted, then a similar process was repeated excluding from two to five measurements at a time. Whenever better results were obtained, the search was restarted, and a single measurement was added from those available that had not yet been considered in the best model. Finally, the entire process was concluded when, after adding and excluding up to five measurements at a time and exhausting all available measurements, a new model that yielded better results was not obtained.

Furthermore, due to the random nature of the measurement search, the entire process described was repeated 1,000 times, keeping only the model that obtained the best results from among the 1,000 created, that is, the one that obtained the highest average coefficient $r_{mean}^*$. This was ultimately called model 1. This model was applied to the $D_{tr}$, $D_{val}$ y $D_{te}$ subsets, and from the BMI predictions, the Pearson correlation coefficients $r_{train}$, $r_{val}$, $r_{mean}^*$ y $r_{test}$ were calculated, as well as their $p_p$ values, when compared with the actual BMI values. Correlation graphs and Bland-Altman graphs were created, and the tract measurements and their respective coefficients of the best BMI quantification model were explicitly reported.

**2.3.2. Model 2: Classification of weight/BMI category.** Model 2 aimed to individually predict the obesity status of subjects by classifying their weight category considering two possibilities: NW and OB, and numerical values equal to -1 and 1 were arbitrarily assigned respectively. The measurements used in the models were also the $N$ chosen in the first part. To create model 2, practically the same process of adding and excluding measurements used for model 1 was followed, with the only differences that instead of correlation coefficients, the precision of the classifications was calculated. Considering the subsets $D_{tr}$, $D_{val}$ and $D_{te}$, the precisions $a_{train}$, $a_{val}$ and $a_{test}$ were obtained respectively. Each one was calculated as the fraction of subjects correctly classified with respect to the total of them in the corresponding subset. To assign a category to the outputs of the models, a threshold equal to 0 was considered such that the outputs lower than the threshold were classified as NW, and higher than the threshold were classified as OB. An average accuracy $a_{mean}^*$ was calculated from the accuracies $a_{train}$ and $a_{val}$, and the condition $a_{mean} > a_{mean}^*$ was tested. The model with the highest $a_{mean}^*$ value was finally called model 2. It was applied to the subsets $D_{tr}$, $D_{val}$ y $D_{te}$, and the accuracies $a_{train}$, $a_{val}$, $a_{mean}^*$ and $a_{test}$ were calculated. Graphs were created to visualize the distribution of classifications. Tract measurements and their respective regression coefficients involved in model 2 were explicitly reported.

**2.3.3. Creation of biomarkers.** Two new obesity biomarkers were proposed. Numerically, each biomarker was constructed from the regression coefficients associated with the tract measurements used in models 1 and 2. Then, each biomarker was reported as a numerical matrix containing the respective coefficients mentioned.

## 3. Results

The tract measurements chosen in the first part are shown in Fig 2. They had significant differences ($p_w < 0.05$) between $D_{norm}$ y $D_{over}$ (condition one) and had significant Spearman correlations ($\rho$) ($p_c < 0.05$, FDR-corrected) with the subjects' BMI when considering $D_{all}$ (condition two). For the tract measurements considered, the $p_w$ values are presented in **Fig 2a**, the $p_c$ values in **Fig 2b** and the $\rho$ correlation coefficients in **Fig 2c**. Positive and negative correlations can be observed in different measurements in **Fig 2c**. Boxplots of the chosen tract measurements for $D_{norm}$ and $D_{over}$ are shown in S1 Fig. Of the 50 tracts studied, only 14 of them met both conditions mentioned using one or more of the 12 measurements. Also, all 12 measurements were considered in at least one tract. Of the 14 tracts, 9 belonged to the brainstem (middle cerebellar peduncle, pontine crossing tract, left and right corticospinal tract, left and right medial lemniscus, left and right inferior cerebellar peduncle, left superior cerebellar peduncle), 2 were projection tracts (left retrolenticular part of internal capsule, left superior corona radiata), and 3 were association tracts (left external capsule, right cingulate gyrus, right inferior fronto-occipital fasciculus), with 8 of them being homologous tracts and 2 central tracts. In the end, the total number of tract measurements studied was $N = 71$ out of a total of 600 possibilities ($= 50$ tracts $\times$ 12 FA measurements).

For comparison, S5 Table lists other papers which also reported correlations between FA values of the 14 tracts considered in the present work and BMI, or differences between NW and OB subjects respectively. S6 and S7 Tables shows more detailed information on the findings of the listed papers, indicating whether the correlations between FA values and BMI were positive or negative.

From the second part of the work, **Fig 3a** shows a graph with the progress of the steps and the number of measurements used for the creation of the models through the process of adding or excluding measurements. A total of 39 steps were completed (indicating the creation of 39 different models, each obtaining better results than the previous one) by adding or excluding measurements. The last model created obtained the highest $r_{mean}^{*}$ value using 41 measurements. **Fig 3b** shows the Pearson correlation coefficients $r_{train}$, $r_{val}$, y $r_{test}$ obtained by applying each of the 39 models created to $D_{tr}$, $D_{val}$ and $D_{te}$, in addition to the average coefficient $r_{mean}^{*}$. Table 2 shows the results of model 1 for predicting BMI values using the 41 tract measurements.

The regression coefficients of model 1 corresponding to the 41 tract measurements are explicitly shown in **Fig 3c**. It can be observed that these measurements included the 14 tracts chosen in the first part of the work. **Fig 3d** shows two matrices, one corresponding to the tract measurements of a subject, and the other to the regression coefficients of model 1 associated with the measurements. By performing the element-by-element product summation between the two matrices, the BMI predicted by model 1 is obtained. Numerically, the matrix with the 41 regression coefficients of model 1 associated with measurements of 14 tracts corresponds to the first obesity biomarker proposed in the present work. The correlation graphs and Bland-Altman plots are shown in Fig 4. These results were obtained after applying model 1 to $D_{tr}$, $D_{val}$ and $D_{te}$, obtaining coefficients of reproducibility (RCP) equal to 6.36, 7.56 and 7.87 kg/m², and coefficients of variation equal to 13%, 16% and 16% respectively.

Similarly to **Fig 3**, the results obtained during the creation of model 2 are shown in Fig 5. **Fig 5a** shows a graph of the process of adding and excluding measurements. Twenty-two steps in total can be observed, indicating the creation of 22 models. The last of these was model 2, which used 28 tract measurements. **Fig 5b** shows the improvement in the average precision $a_{mean}^{*}$ during the steps. Also, in **Fig 5c** the regression coefficients of model 2 associated with the 28 tract measurements used are explicitly shown. It can be observed that these

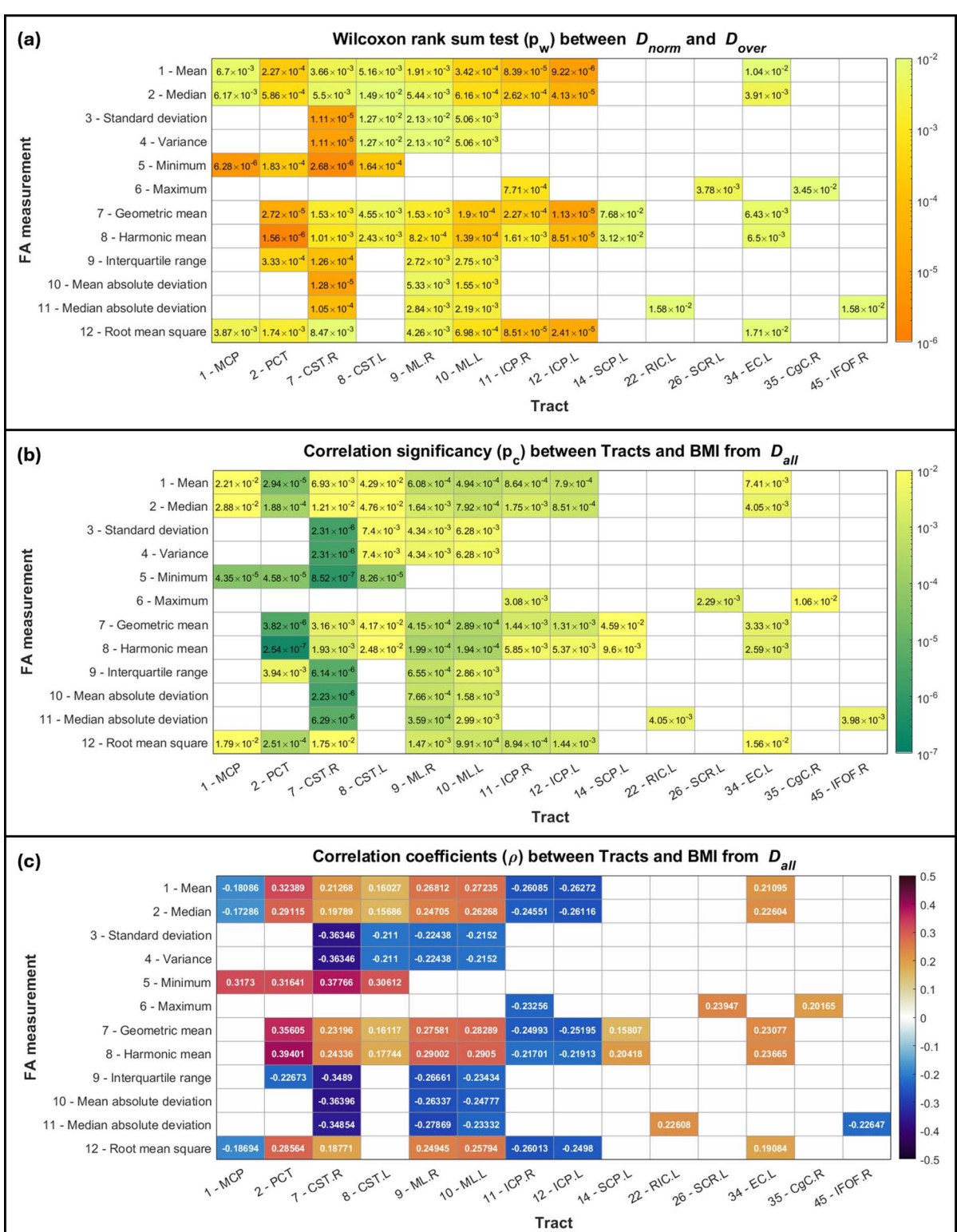

**Fig 2. Tracts and their measurements that met the two conditions of the first part of the study ($p_w < 0.05$ and $p_c < 0.05$, FDR-corrected).** (a) Comparing the data between $D_{norm}$ y $D_{over}$ by applying the Wilcoxon rank sum test, the $p_w$ values of 14 tracts and their respective measurements that met the two conditions are shown. In total, 71 tract measurements were chosen. (b) $p_c$ values indicating the significance ($p_c < 0.05$, FDR-corrected) of the Spearman correlations between the 71 tract measurements and the BMI of the subjects considering $D_{all}$. (c) Spearman correlation coefficients ($\rho$) of the previous tract measurements.

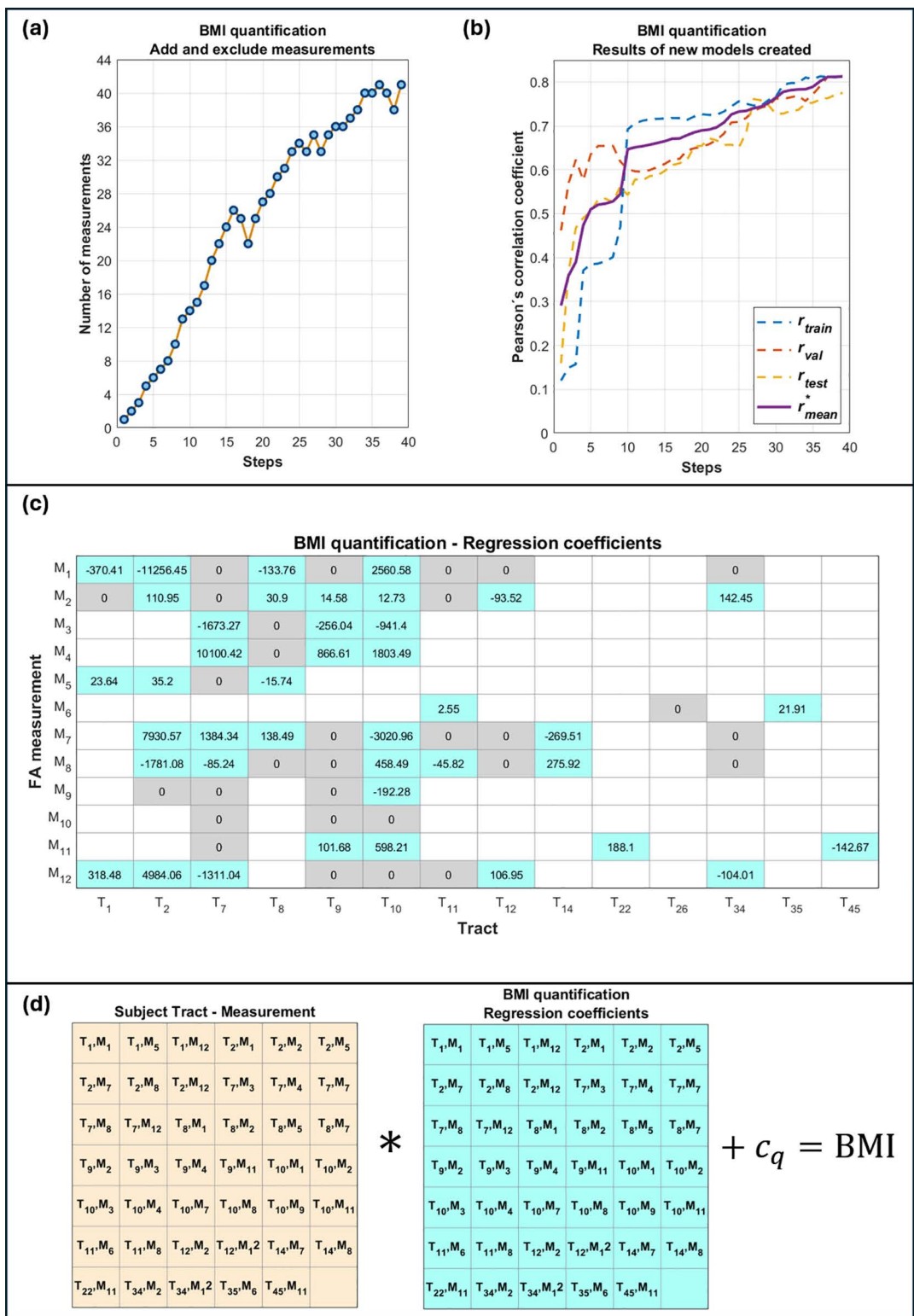

**Fig 3. Results of the process of creating model 1 to quantify BMI.** (a) Number of measurements used and (b) correlation coefficients $r_{train}$, $r_{val}$, $r_{test}$ y $r_{mean}^{*}$ from each step. (c) Correlation coefficients obtained for model 1, associated with 41 tract measurements (blue cells). (d) Two matrices are shown, one containing the tract measurements of some subject, and another containing the regression coefficients of model 1 shown in (c). Performing the element-wise product summation of the two matrices and adding the model constant $c_q$ = 142.87 kg/m², the BMI predicted can be obtained.

**Table 2. Results of model 1 for predicting BMI.** The results obtained after applying the model to predict BMI to the $D_{tr}$, $D_{val}$ and $D_{te}$ subsets are shown, indicating the Pearson correlation coefficient, the 95% confidence interval and the $p_p$ value of the correlation indicating its significance. Their respective root mean square error (RMSE) and mean absolute percentage error (MAPE), for the subjects of $D_{tr}$, $D_{val}$ and $D_{te}$ are presented in the last columns of this table.

| Subset | Pearson correlation coefficient | 95% confidence interval | $p_p$ | RMSE | MAPE |
|---|---|---|---|---|---|
| Training ($D_{tr}$) | 0.8124 | (0.7215, 0.8758) | $5.89 \times 10^{-20}$ | $3.22 \, kg/m^2$ | 9.92% |
| Validation ($D_{val}$) | 0.8130 | (0.6716, 0.8973) | $1.85 \times 10^{-10}$ | $4.18 \, kg/m^2$ | 13.92% |
| Testing ($D_{te}$) | 0.7753 | (0.6116, 0.8754) | $4.23 \times 10^{-9}$ | $3.99 \, kg/m^2$ | 12.6% |

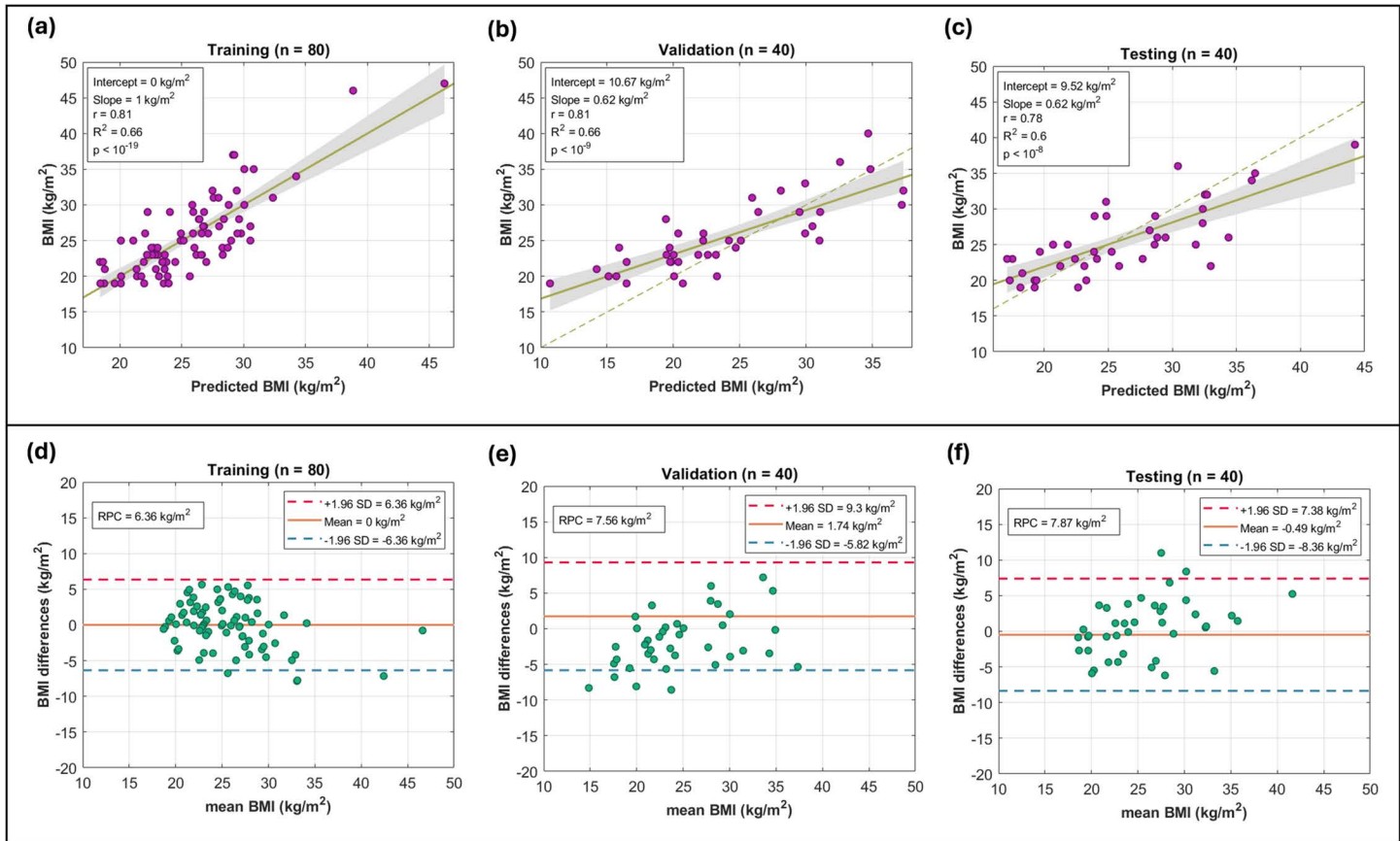

**Fig 4. Correlation and Bland-Altman plots.** Correlation plots are shown in parts (a), (b), and (c), indicating for each subject in the training, validation, and testing subsets ($D_{tr}$, $D_{val}$ and $D_{te}$, respectively), the actual BMI values and those predicted by model 1. Bland-Altman plots are shown in parts (b), (d), and (f). Reproducibility coefficients (RPC), mean and $\pm$ 1.96 times the standard deviation (SD) are indicated.

measurements included 12 of the 14 tracts chosen in the first part (ceasing to consider left superior corona radiata and right inferior fronto-occipital fasciculus). **Fig 5d** shows a diagram of two matrices, one containing the 28 tract measurements of a subject, and the other the correlation coefficients of model 2. The operation between them allowed to classify the weight category of a subject. Numerically, the matrix with the 28 regression coefficients of model 2 associated with measurements of 12 tracts corresponds to the second obesity biomarker proposal in the present work. Fig 6 shows graphs with the distribution of the predictions of model 2. Table 3 shows the results of sensitivity, specificity and accuracy for $D_{tr}$, $D_{val}$ and $D_{te}$, in addition to their root mean square error (RMSE) and mean absolute percentage error (MAPE)

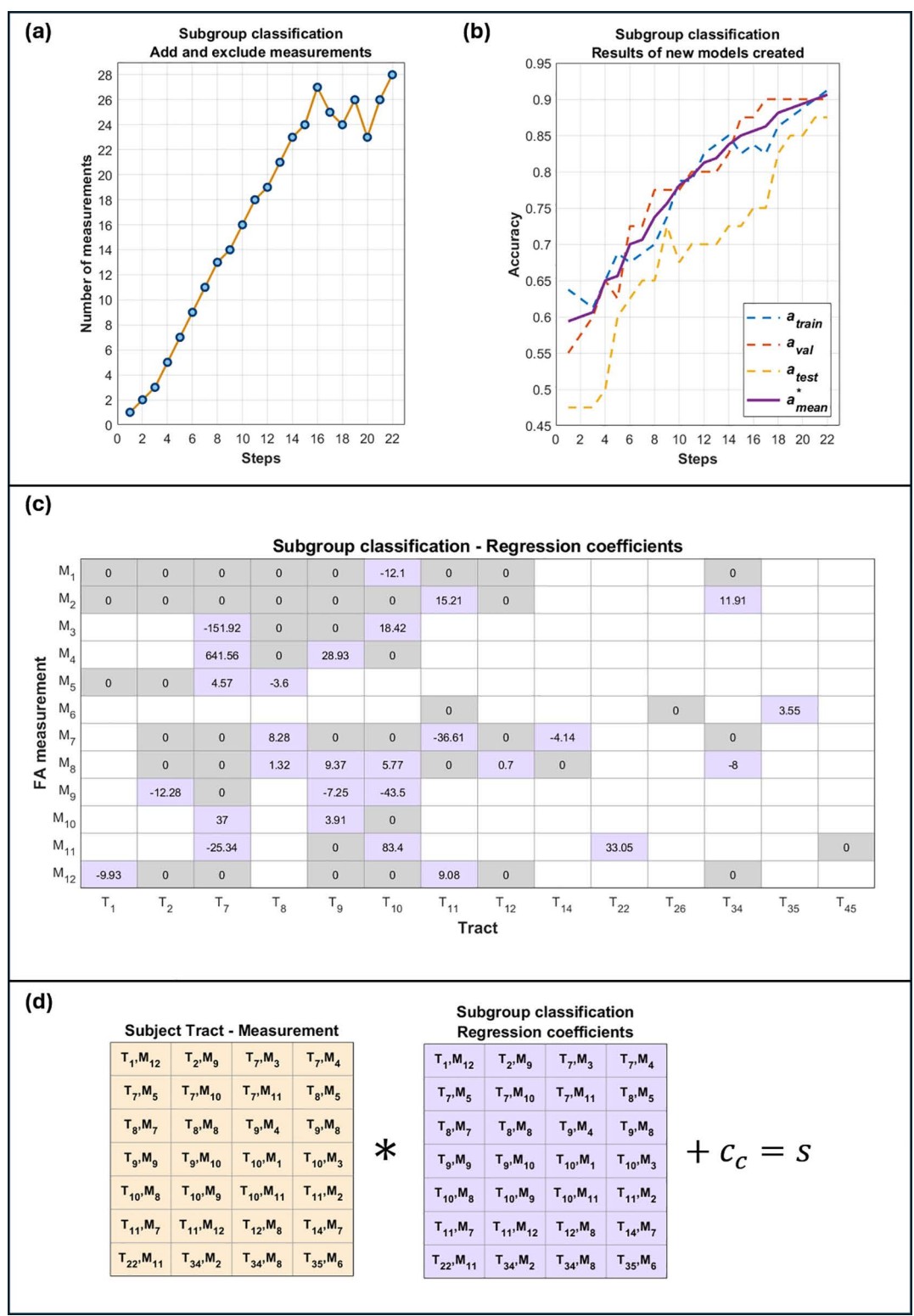

**Fig 5. Results of the process of creating model 2 to classify the weight category of the subjects.** (a) Number of measurements and (b) accuracies $a_{train}$, $a_{val}$, $a_{test}$ and $a_{mean}^*$ from each step. (c) Regression coefficients obtained for the model 2, associated with 28 tracts (purple cells). (d) Two matrices are shown, one containing the tract measurements of some subject, and another containing the regression coefficients of model 2 shown in (c). Performing the element-wise product summation of the two matrices and adding the model constant $c_c = 5.69$, the output $s$ can be obtained to classify the subject considered.

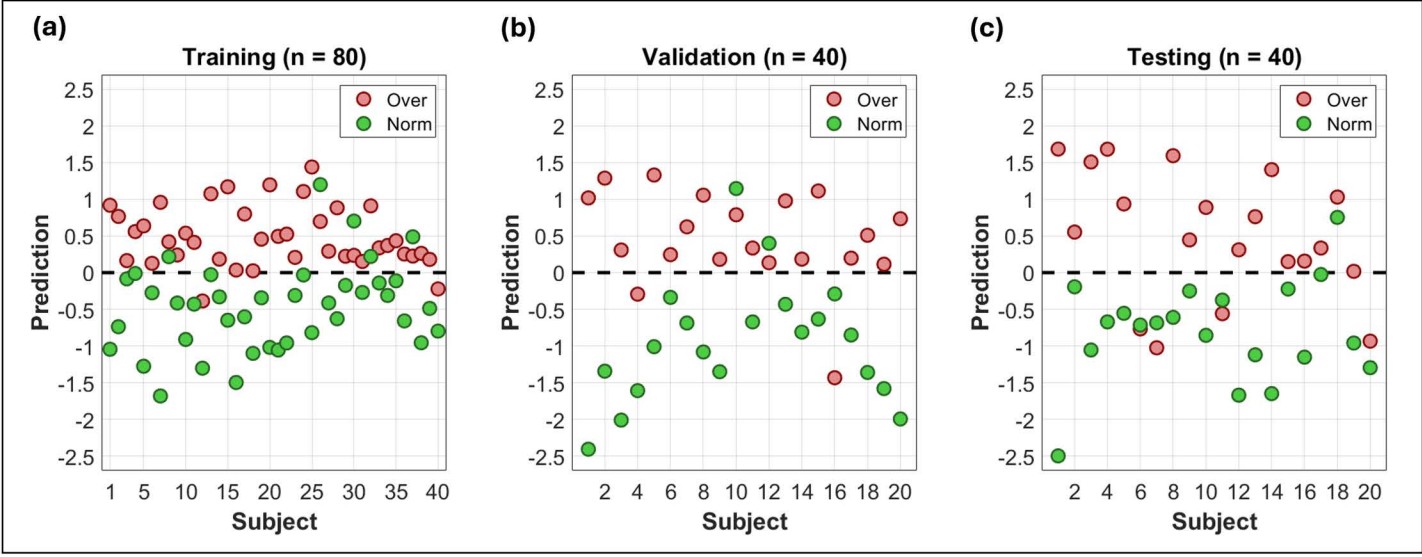

**Fig 6. Plots of classification results.** The distributions of the $s$ outputs from model 2 applied to (a) $D_{tr}$, (b) $D_{val}$ and (c) $D_{te}$ respectively are shown. Green circles correspond to subjects with actual normal weight, and red circles to those with actual overweight/obese. A dotted black line at $y = 0$ indicated the threshold against which subjects were classified. For a subject, if his output was $s < 0$, then that was classified as NW; and if it was $s > 0$, then that was classified as OB.

**Table 3. Results of model 2 for subject classification.** Sensitivity, specificity, and accuracy values are shown, along with their respective root mean square error (RMSE) and mean absolute percentage error (MAPE), for the subjects of $D_{tr}$, $D_{val}$ and $D_{te}$.

| Subgroup | Sensibility | RMSE | MAPE | Specificity | RMSE | MAPE | Accuracy | RMSE | MAPE |
|---|---|---|---|---|---|---|---|---|---|
| Training ($D_{tr}$) | 0.875 | 0.707 | 25% | 0.95 | 0.447 | 10% | 0.9125 | 0.591 | 17.5% |
| Validation ($D_{val}$) | 0.9 | 0.633 | 20% | 0.9 | 0.633 | 20% | 0.9 | 0.633 | 20% |
| Testing ($D_{te}$) | 0.95 | 0.447 | 10% | 0.8 | 0.895 | 40% | 0.875 | 0.707 | 25% |

values. The algorithms developed in this work, in addition to the data and results generated, can be freely downloaded for reproducibility in [27].

## 4. Discussion

After studying two sets of subjects (NW and OB), formed only by young adult women with a medium level of education; results obtained were opposite to what is usually reported in the literature. These consisted of positive correlations between WM integrity and obesity mainly in tracts involved in motor functions. Besides, a process for the selection of tract measurements was proposed, such that precise models were built to quantify and classify the obesity level or status of subjects individually. From these, novel brain biomarkers of obesity were proposed, formed by the regression coefficients involved in the models.

### 4.1. WM tract measurements

In this work, the WM tracts were studied through the calculation of descriptive statistics on FA images. All statistics used are well known, although some of them are not usually used to characterize information coming from the WM tracts. The most common statistics used here were the arithmetic mean, median, standard deviation and range [28–32]. In addition to the usual ones, the variance, the harmonic mean, the geometric mean, the interquartile

range, maximum, minimum, media absolute deviation, median absolute deviation, and the root of the square mean were used, giving a total of 12 statistics. Particularly, in the case of the harmonic mean and the geometric mean, it is known that they are useful for data sets with specific characteristics [33]. The harmonic mean is used instead of the arithmetic mean, when data are expressed as ratios or ranges on different scales. In the case of the geometric mean, it is convenient to use it when there are multiplicative relationships between the data (for example, compound interests). Thus, these expressions of the mean value are used in special situations so that the value obtained is closer to the desired "average." However, there is no explicit prohibition for the use of these statistics when studying data that do not necessarily have the aforementioned characteristics. Furthermore, the geometric and harmonic means were not used looking for a "more correct" average value, but rather they were only used to characterize information from the tracts in a different way. This was also the same reason for using the other statistics already mentioned. As could be seen in **Fig 3b**, the central tendency statistics, that is, the (arithmetic) mean, median, geometric mean and harmonic mean, had in common that for the same tract, they all had a significant correlation (positive or negative). On the other hand, based on their definitions (S4 Table), there was no linear relationship between them, so although they may represent similar information, they were independent and complementary during the creation of the models 1 and 2. The information extracted calculating the statistics on the FA images proved to be useful to achieve the objectives of the work, since they were part of the selected tract measurements in the models 1 and 2.

## 4.2.  Positive correlations between WM integrity and BMI

Many works have reported reduced WM integrity in OB subjects compared to their NW peers. This has been presented as a negative correlation between FA measurements from different tracts and obesity-related measures, such as BMI, waist circumference, fat percentage, etc. [5,7,8]. Considering only tract measurements using central tendency statistics (to be able to make comparisons with those obtained by other works), among the 14 reported tracts, 10 of them showed significant correlations with BMI considering the arithmetic mean, median, geometric mean and/or harmonic mean (**Fig 2c**). Among these, 9 major white matter tracts belonged to the brainstem and were involved in motor functions (S4 Table). From these, 6 showed positive correlations between FA measurements and BMI (pontine crossing tract, left and right corticospinal tracts, left and right medial lemniscus, and left superior cerebellar peduncle). Corticospinal tracts are the major neuronal pathways providing voluntary motor function. They are formed by descending fibers connecting the motor area with the spinal cord to enable distal limb movement [34]. In a work reported by Lv et al. [35], the authors analyzed the effect of cumulative BMI over 16 years on neuroimaging features of brain health in adults of different ages (25–83 years). TBSS and genetic analysis were conducted to test differences between age groups. WM integrity at the voxel level demonstrated a positive association between cumulative BMI and FA in the pontine crossing tract, middle cerebellar peduncle, and projection fibers (bilateral corticospinal tract). Differences among age groups were only significant in young adults (age < 45 years). Based on their findings, the authors suggested that high BMI may be associated with adaptive neuroplasticity or selective neurodegeneration, resulting in higher FA values of projection fibers. The authors based their hypothesis on the work of Fu et al. [36] in which plasticity of corticospinal tracts was demonstrated during the early stages of Parkinson's disease (PD). In that work, longitudinal diffusion tensor imaging before and after the development of prodromal motor PD showed higher FA in corticospinal tract compared to controls, indicating adaptive structural changes in motor networks in concert with nigrostriatal dopamine, as a compensatory process due to the loss of motor

skill control. Besides, activity-dependent neuroplasticity of the corticospinal system has been also reported, for example, during rehabilitation after stroke, where segments in the ipsilesional corticospinal tract showed higher FA values one year after the start of rehabilitation [37]. Although neuroplasticity can occur simultaneously in different parts of the brain, in the aforementioned works, the authors reported adaptation processes in specific, non-dispersed WM tracts during their longitudinal studies. In the present work, the data studied represented a single instant in time, with the consequence that no result can imply adaptation processes related to changes in the subjects' obesity status or to changes in WM integrity. However, it was interesting to find positive correlations between the subjects' BMI and the integrity of WM tracts related to motor functions, as reported by previous works. On the other hand, it should be remembered that, unlike others, in the present work only young adult women were studied. A study that looked for differences in neuroplasticity between females and males along the cortico-spinal tract and the corpus callosum, but without considering any obesity measure as a covariate, as reported by Kirby et al. [12]. Here, authors examined older adults before and after visuo-motor training using FA and myelin water fraction. Among their results, they found significant FA increase in the cortico-spinal tract in females and not males. In another work, Ho et al. [38] studied a sample of young male and female adolescents by examining associations between longitudinal changes in gonadal hormones and corresponding changes in several WM tracts. Authors found positive associations between changes in testosterone and increased FA within the corpus callosum, cingulum cingulate, and corticospinal tract in females, but non-significant associations in males.

A well-established fact is that increased BMI is associated with higher testosterone levels in women and decreased levels in men [39]. From the cited works, in women higher testosterone levels were associated with greater integrity of corticospinal tracts, and a higher BMI was associated with higher testosterone levels. Therefore, a hypothesis is that in women, higher BMI was associated with greater integrity in corticospinal tracts, which also coincides with the results obtained for the present work. Furthermore, if positive correlations found between FA measurements and BMI in the bilateral corticospinal tract and medial lemniscus, as well as the pontine crossing tract, were the result of adaptive processes in WM, it could be suggested as a hypothesis that increased FA values in WM tracts associated with motor functions could be related to a greater effort to perform voluntary movements of the limbs as a consequence of the difficulty of handling a greater weight. However, it should be emphasized that these hypotheses are only speculative, due to the differences between the methodologies used in previous works and the age ranges studied here. In addition, in the present study, only women and not men were studied, so it was not possible to ensure that the results were exclusive to women.

The positive correlations obtained in this work differ from those usually reported by others, where negative correlations between WM integrity and obesity measurements were obtained. These differences could have been due to the ages of the subjects studied. In this work the age range was restricted from 19 to 26 years, thus being all young adults, to minimize the effect of aging on any brain-related measurements. In other works, however, subjects with a wider age range have been studied, including from young adulthood to adults at older ages. To date and to the best of our knowledge, only three studies have reported a positive correlation between any measure of obesity and WM integrity [10,11,14]. Thus, then the results reported in this work complemented and expanded the existing evidence on the WM integrity in obesity. Finally, it can be concluded that the first hypothesis of the study was fulfilled, since comparing NW and OB and studying only young adult women, resulting in opposite findings to those usually reported.

### 4.3. Quantification and classification models

Based on information from WM tracts, few studies have proposed models to predict the obesity status of subjects individually, either by quantifying the BMI or classifying their weight category [2,3,15,16]. Obviously, it is not useful to quantify or classify the BMI of a subject through complicated and sophisticated methodologies, when this is easily done with conventional instruments. The objective of the models goes in the opposite direction, that is, if a set of tract measurements can predict the obesity status of a subject, then it is hypothesized that these measurements are closely linked to what characterizes the obesity status of the subject. This is intended to provide new knowledge about the effects that obesity causes on WM. In the present work, models 1 and 2 were created to perform the tasks of quantifying the BMI and predicting the weight category respectively. The tract measurements initially considered for their construction were those previously chosen in the first part of the work as they met two conditions: one, having presented significant differences between the subjects that formed $D_{norm}$ and $D_{over}$; and two, to present significant correlations with the subjects' BMI considering $D_{all}$. The tract measurements that ultimately built models 1 and 2 subsequently allowed the proposal of two new biomarkers associated with obesity.

Among the works that have attempted to quantify or classify BMI based on brain characteristics, is the one reported by Byeon et al. [3], who proposed an improvement of the method known as functional correlation tensor, incorporating T1-weighted spatial information. Using the FA images, through the LASSO framework, they identified 26 ROIs from major fiber bundles to predict BMI and to perform classification into three weight subgroups. Predicted BMI showed a mean correlation equal to 0.57, with mean RMSE equal to $4.96 \pm 0.65$. For classification they obtained an average precision equal to 57.31%. In other work, Park et al. [15] explored a multi-modal approach to predict BMI through connectivity analysis. Significant regions and associated imaging features were identified based on group-wise differences. Using a partial least-square regression (PLSR) framework, their model obtained a correlation coefficient equal to 0.4414, an RMSE equal to 5.26 and SD equal to 5.26. Okudzhava et al. [2] used connectome-based predictive modeling to predict BMI. Multiple linear regression models were built and a leave-one-out cross-validation was used. Their model obtained a correlation coefficient equal to 0.46. Vakli et al. [16] used a convolutional neural network (CNN) for BMI prediction based on T1-weighted structural MRI of the whole brain. The CNN had 230,961 trainable parameters. They used transfer learning to investigate the generalizability of their approach, adapting the model to a different dataset (Images (IXI) dataset). They applied Gradient-weighted Class Activation Mapping (Grad-CAM) to localize brain regions that made a significant contribution to BMI prediction. For the IXI dataset, MAE = $3.00 \, \text{kg/m}^2$, STDAE = $2.12 \, \text{kg/m}^2$, RMSE = $3.67 \, \text{kg/m}^2$, Pearson r = 0.49, and $R^2$ = 0.21 were obtained.

Therefore, simple models from linear regressions to more complex ones using neural networks have been used to predict obesity status of individual subjects. Simpler models facilitate interpretation, allowing the generation of hypotheses about why measurements in the models were associated with what they predicted or classified. More complex models make this task more difficult, since there are a larger number of variables involved that are not trivial to interpret or associate. To reduce the complexity of the models due to the number of variables involved, different processes for model reduction and simplification have been used. In the present work, a simple and original process was proposed to perform this task.

### 4.4. Process of adding and excluding measurements

In the second part of the work, a process was proposed to progressively add or exclude from 1 to 5 tract measurements at a time to create new and better models. From this

process, models 1 and 2 were built to quantify and classify the level or state of obesity of subjects individually, respectively. Other works that proposed models for BMI quantification and/or classification of subjects according to their weight category used other complex methods for the selection of variables. For example, Byeon et al. [3] used a LASSO framework, which was a regularized regression analysis to select a sparse set of variables that could explain a dependent variable. Park et al. [15], used a partial least-square regression (PLSR) framework, which was a combination of principal component analysis (PCA) and multiple linear regressions. Gupta et al. [40] used sparse partial least squares for discrimination analysis (sPLS-DA). sPLS simultaneously performed variable selection and classification using LASSO penalization. In addition, variable importance in projection (VIP) scores were calculated and a stability analysis was used. It is worth mentioning that there are other proposed models that did not require a variable selection process. Examples of these are the works of Vakli et al. [16] and Finkelstein et al. [41]. In both cases, the models were created from the training, validation and testing of CNNs. It is known that CNNs are specialized for image analysis and have the advantage of automatically learning the necessary image features, without the need of user input. On the other hand, a CNN is characterized by involving a large number of trainable parameters (for example, a total of 230,961 in the work of Vakli et al.), having the characteristic that they cannot have a direct interpretation as in regression models that involve a set of well-identified variables. Since their parameters were not interpretable, other methods have been used to identify the brain regions that contributed most to CNNs for prediction and/or classification tasks. Among them is the Grad-CAM method used by Vakli et al. or the explainable AI (XAI) maps used by Finkelstein et al., which produced localization maps that highlighted regions in the input image (in this case, obtained from the brain) that were important for prediction and/or classification.

Unlike other works, the proposed process for measurement selection can be partially considered as a "brute force algorithm", since it was a simple and straightforward approach in which almost all possible combinations were tested randomly. However, for model 1, adding the condition $r_{mean} > r_{mean}^*$, reduced the search space and improved efficiency. The methodologies for variable selection in other works are mainly useful when there is a large number of features, so overfitting is desired to be avoided by reducing the complexity of the model. In our case, the number of possible features was relatively low. Considering initially 50 tracts and 14 statistical measurements, 600 tract measurements were available. Then, in the first part, applying the two required conditions, this number was reduced to 71. From this number of measurements, the simple and novel process described for choosing variables was applied, and two models were created, one to quantify BMI using 42 tract measurements (from 11 statistics measured in some of 13 tracts), and another for classification using 28 tract measurements (from 12 statistics measured in some of 12 tracts). Therefore, the proposed models were characterized by using a relatively small number of measurements before and after the measurement selection process. In addition, it required a small amount of computational resources during its application, since it started from one measurement onwards, while the other methodologies mentioned started from the total number of measurements and advanced by reducing the number of them by applying more complex and demanding processes from the beginning. Although other reported methods were more complex and statistically robust, in the present work the results obtained using the proposed process allowed obtaining high correlations with the real BMI values by applying model 1, and high precision in the classification of weight categories by applying model 2, thus demonstrating its effective and convenient utility for the problem addressed.

### 4.5. Brain biomarkers associated with obesity

Brain biomarkers allow the evaluation of individual information extracted from multimodal analysis methodologies. This is useful as clinical tools to help with the diagnosis, monitoring and prognosis of diseases, response to drugs, or otherwise, to establish objective and quantitative associations between differential characteristics and study groups [42,43]. Therefore, these biomarkers have potential clinical applications, in addition to being useful for the generation of new knowledge that allows directing novel research based on their findings. Among the most studied brain biomarkers are those that describe the difference between chronological age and predicted age based on structural and functional neuroimaging data. These have been associated with neurodegenerative diseases, such as schizophrenia, Alzheimer's and Parkinson's diseases, multiple sclerosis, or with an increase in patient mortality [44–46]. Regarding brain biomarkers associated with obesity, there are those studying quantitative MRI metrics relating obesity with neuroinflammation in brain regions involved in food intake [47], with negative emotional states [48], or with dysfunction in executive regions causing cognitive deficit [49]. In the present work, two novel biomarkers that associated obesity with decreased or increased integrity of several WM tracts were proposed. Regarding the tracts that presented increased integrity, it is known that these are related to motor functions. The proposed biomarkers were represented as fixed value matrices, formed by the regression coefficients of models 1 and 2, and based on well-known but rarely used statistical measurements to characterize WM integrity. These biomarkers were interesting findings, since part of them were constructed from information on an increased integrity of motor tracts, that differ from what is usually reported by other works. The possible main reason for such differences lies in the specific demographic characteristics of the study group formed by young right-handed women with a medium level of education. This was different from what is usually studied by other works, in which they analyze wide ranges of ages and mixed sexes with the objective of generalizing their results. Considering all the above, it can be concluded that the second hypothesis of the work was fulfilled, because it was possible to develop novel biomarkers associated with obesity in young adult women and built from tract measurements involved in the predicting models. Still, the final objective of the present work was that the findings had an impact on clinical applications for the identification of risk factors associated with brain alterations, or new mechanisms of action for the prevention and treatment of these changes [7]. These biomarkers were an initial suggestion that sought to direct future research based on the reported findings. After validation by independent works studying larger samples, the usefulness of these biomarkers associated with obesity can be validated.

### 4.6. Limitations

Limitations of the present work include the small sample size and the lack of generalizability of the results due to the restriction of demographic characteristics, that is, having studied only right-handed young adult women with medium level of education. On the other hand, the type of analysis performed (VBA) has known methodological limitations, particularly the insufficient specificity of the measured parameters and the limited accuracy of WM tract reconstruction, because DTI has little capacity to differentiate crossing fibers within a voxel [9]. This may consequently imply that FA alterations may reflect different underlying factors such as axonal loss or injury, changes in myelin content, inflammation, or shifts in extracellular and intracellular water concentrations [2]. Also, there is always a limitation in correlation studies on the causal interpretation of the results, since it cannot be concluded whether the brain alterations are caused by obesity, or whether obesity causes those brain alterations. Interpreting findings in a biological context remains a challenge.

In this work, BMI was the only measurement associated with obesity. Considering that this index is more sensitive to the lean mass of the subjects than to their body and abdominal fat content, other measurements associated with adiposity should be used. These could include waist measurement, percentage of adipose tissue, amount of subcutaneous fat, amount of visceral fat, among others. The proposed biomarkers should also be applied to other databases to validate the results in women and extend them to men with different ages. Only then could the results imply a possible generalization. Therefore, there is a need to conduct further work in which differences between women and men are studied, with different age ranges, in addition to doing longitudinal studies. The latter could allow for the analysis of possible adaptation and neuroplasticity processes in WM tracts related to motor functions reported in other works as well as in the present one.

The use of a novel and simple process for the selection of variables for regression models was proposed, so that other statistically more robust methods to identify multicollinearities between variables could be implemented (such as the well-known LASSO and Ridge regressions). Also, stability-based quantification metrics could be considered based on simple cross-validations. Finally, in addition to multiple linear regression models, other models can be considered based on CNNs, combined with novel interpretation methods to identify those features that have a greater contribution to the prediction models.

## 5. Conclusions

Novel brain biomarkers associated with obesity were sought by studying tract measurements based on well-known but not commonly used statistics to characterize WM tracts. In the present work, subjects with specific demographic variables were studied, such as being right-handed, women aged between 19 and 26 years, and with a medium level of education. This was done to avoid variations in brain measurements dependent on other confounds such as age and gender. After choosing tract measurements that showed significant differences between the two groups formed by subjects with NW and OB respectively, and whose measurements were also correlated with the subjects' BMI, a simple and novel process was applied to choose the measurements that would allow the creation of models to quantify and classify the obesity status of individual subjects. The biomarkers sought were then constructed from the regression coefficients of the models. Among the tract measurements involved in the models, several were found to show positive correlations between WM integrity and obesity, mainly in tracts related to motor functions. This was contrary to what is usually reported. As future work, the process carried out should be repeated in larger databases, including men, as well as a wider age range, to obtain results that can be generalized. Finally, the proposed novel obesity biomarkers allowed the generation of new knowledge and have the ultimate goal of subsequently being a useful tool in the clinical environment for the prevention and treatment of WM changes due to obesity.

## Supporting Information

**S1 Fig. Boxplots of the tract measurements studied.** Boxplots for the Dnorm (blue) and Dover (red) sets, with the white matter tract indicated on the horizontal axis and the value of the measured statistic on the vertical axis. Only the tracts and measurements that had statistically significant differences ($p_w < 0.05$) after applying the nonparametric Wilcoxon rank sum test, comparing the subjects of the $D_{norm}$ and $D_{over}$ sets, are presented. These tracts and measurements are the same ones that obtained significant Spearman correlations ($p_c < 0.05$, FDR-corrected) with the BMI values of all the study subjects of the $D_{all}$ set, which can be observed in Fig 2.
(PDF)

**S2 Fig. Distribution of subjects by subset and BMI** . For the training, validation and testing subsets, 80, 40 and 40 subjects respectively were randomly selected. An approximate 2:1 ratio was sought between subjects with the same BMI value in the training subset and the validation or testing subsets.
(PDF)

**S1 Table. Subjects with normal weight studied in this work.** Participant IDs provided in the ID1000 database.
(PDF)

**S2 Table. Subjects with overweight/obesity studied in this work.** Participant IDs provided in the ID1000 database.
(PDF)

**S3 Table. ICBM-DTI-81 white-matter labels atlas.** White matter parcellation.
(PDF)

**S4 Table. Tract measurements.** For each tract, 12 descriptive statistics were calculated from the voxel values considering the FA images.
(PDF)

**S5 Table. State of the art of WM integrity vs. Obesity.** Details on published papers that have reported correlations between WM integrity and obesity.
(PDF)

**S6 Table. Comparison of findings of this work vs. state of the art.** The tracts that presented significant correlations ($p_c < 0.05$, FDR-corrected) in this work, between WM integrity and BMI, are listed. The statistics mean (arithmetic), median, geometric mean and/or harmonic mean calculated on the tracts in the FA images were considered. Correlations are presented as a color scheme, positive values (blue cells) or negative values (pink cells) were used to this end. Works that reported findings in the listed tracts are also highlighted. Abbreviations used: N: normal weight; OV/OB: overweight/obese; BMI: body mass index; AFR: abdominal fat ratio; WC: waist circumference.
(PDF)

**S7 Table. Comparison of findings of this work with state of the art that used other statistics.** Data is presented in the same manner as S4 Table. Abbreviations used: N: normal weight; OV/OB: overweight/obese; BMI: body mass index; AFR: abdominal fat ratio; WC: waist circunference; BFP: body fat percent.
(PDF)

## Author contributions

**Conceptualization:** José Gerardo Suárez-García, Martín Rodolfo Palomino-Merino, Benito de Celis-Alonso.

**Data curation:** Benito de Celis-Alonso.

**Formal analysis:** José Gerardo Suárez-García, Benito de Celis-Alonso.

**Funding acquisition:** Benito de Celis-Alonso.

**Investigation:** José Gerardo Suárez-García, María Isabel Antonio-de la Rosa, Nora Coral Soriano-Becerril, Javier M. Hernández López.

**Methodology:** José Gerardo Suárez-García, Martín Rodolfo Palomino-Merino.

**Project administration:** Benito de Celis-Alonso.

**Resources:** Benito de Celis-Alonso.

**Software:** José Gerardo Suárez-García, María Isabel Antonio-de la Rosa, Nora Coral Soriano-Becerril, Javier M. Hernández López.

**Supervision:** Martín Rodolfo Palomino-Merino, Benito de Celis-Alonso.

**Validation:** José Gerardo Suárez-García.

**Writing – original draft:** José Gerardo Suárez-García.

**Writing – review & editing:** José Gerardo Suárez-García, María Isabel Antonio-de la Rosa, Nora Coral Soriano-Becerril, Javier M. Hernández López, Martín Rodolfo Palomino-Merino, Benito de Celis-Alonso.

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
