## [Decision Letter · Decision Letter 0]

28 Oct 2024

PONE-D-24-39635Novel brain biomarkers of obesity based on statistical measurements of white matter tractsPLOS ONE

Dear Dr. de Celis-Alonso,

Thank you for submitting your manuscript to PLOS ONE. After careful consideration, we feel that it has merit but does not fully meet PLOS ONE’s publication criteria as it currently stands. Therefore, we invite you to submit a revised version of the manuscript that addresses the points raised during the review process.

**ACADEMIC EDITOR: **

Based on the reviewers’ comments, the manuscript requires extensive revision to meet the publication standards of *PLOS ONE* . Below is an outline of the required changes for acceptance, recommended improvements, and additional feedback, structured to address any conflicting advice and enhance clarity on the actions needed.

<h3>Required Changes for Acceptance</h3>

**Clarification and Definition of Acronyms****Required Change** : Define acronyms (e.g., NW for Normal Weight, OB for Obese) upon first use in the text (line 132), not after (line 136). Clear terminology improves readability and consistency for the reader.
**Consistency in Software Use and Image Orientation****Required Change** : Address potential inconsistencies caused by using two different software tools (MRtrix3 and FSL). Perform a validation check as outlined by Reviewer 2 by testing the analysis with one subject, verifying that the gradient table and image orientations (Left-Right, Anterior-Posterior, Superior-Inferior) are consistent across all data processing steps. This is critical to ensure reliability in results and should be reported in the manuscript.**Conflict Resolution** : If following only one software is not feasible, explicitly state the reasons and provide detailed explanations on how consistency across outputs was maintained and validated.
**Hypothesis Specification****Required Change** : Clarify the hypotheses regarding which white matter tracts may be associated with obesity and justify these choices based on relevant literature. The current “shotgun” approach should be refined to a more targeted hypothesis that strengthens the theoretical basis of the study.**Conflict Resolution** : Align the hypothesis with the stated goal of understanding the relationship between specific white matter tracts and obesity.
**Sample Generalizability****Required Change** : Explicitly discuss the limitations of the sample (i.e., young, healthy females aged 19-26) and address how these constraints affect generalizability. Include a more precise description of how this sample is representative (or not) and clarify that the findings are specific to this demographic subset.**Conflict Resolution** : Ensure that any generalization to broader populations is cautious and adequately qualified to avoid overstatement.
**Neuroplasticity Discussion****Required Change** : Avoid broad statements about neuroplasticity being “rampant” and instead provide a more detailed analysis of why only certain white matter tracts might be associated with obesity. This analysis should align with the specific tracts examined and avoid implying unsupported causation.**Conflict Resolution** : Ensure statements about neuroplasticity are focused on observed associations within the study’s constraints and are not overly generalized.
**Language and Consistency****Required Change** : Address the manuscript’s language issues, including grammar, spelling, tense consistency, and voice. Ensure the manuscript is written in either first-person or passive voice consistently, not a mix of both.**Conflict Resolution** : This is a crucial change for clarity and professionalism, as inconsistent language can undermine the study’s perceived rigor and readability.

<h3>Recommended Improvements (Not Required for Acceptance)</h3>

**Data on Eddy Current Distortion Management****Recommended Improvement** : Detail how high signals from eddy current distortions were managed in the FA images (e.g., removing bright voxels at the edges of the brain). While not critical, explaining these steps will enhance the transparency and quality of data processing.
**Refining the Neuroplasticity Interpretation****Recommended Improvement** : Provide a more nuanced discussion of neuroplasticity to address why only specific tracts may be associated with obesity, rather than assuming associations are widespread. This will improve the discussion’s scientific precision.
**Spelling and Grammar in Depth****Recommended Improvement** : Conduct an additional layer of proofreading focused on minor language issues (e.g., typos, tense inconsistencies). This will improve readability but is supplementary to the broader language consistency required above.

<h3>Specific Feedback on Manuscript Sections</h3>

**Abstract and Introduction**Ensure hypotheses are clearly stated and aligned with the study objectives. Define abbreviations when first used to enhance clarity.
**Methods**Provide a thorough description of all software used, including validation of consistency in data processing steps if using multiple software tools.Specify how high-signal voxels from eddy current distortions were handled.Detail the justification for specific white matter tracts examined based on theoretical or empirical grounding to align with the study’s objectives.
**Results**Ensure all results reported strictly relate to the statistical analyses stated in the Methods section.Clarify the rationale for significant differences across white matter tracts and link these findings back to the hypotheses.
**Discussion**Ensure the neuroplasticity findings are discussed with nuance, avoiding generalizations across the brain and focusing instead on specific tracts related to obesity.Discuss the limitations of the study sample in terms of generalizability, clearly outlining how these limitations may affect broader applications of the findings.Ensure that interpretation statements, especially on neuroplasticity, are well-supported by the study’s data.
**Conclusion**Avoid overstating the findings’ generalizability to broader populations or implying causation where only associations were observed.

We look forward to receiving your revised manuscript.

Kind regards,

Maryam Bemanalizadeh

Academic Editor

PLOS ONE

2. Please note that PLOS ONE has spec6ific guidelines on code sharing for submissions in which author-generated code underpins the findings in the manuscript. In these cases, all author-generated code must be made available without restrictions upon publication of the work. Please review our guidelines at https://journals.plos.org/plosone/s/materials-and-software-sharing#loc-sharing-code and ensure that your code is shared in a way that follows best practice and facilitates reproducibility and reuse.

Reviewers' comments:

Reviewer's Responses to Questions

**Comments to the Author**

1. Is the manuscript technically sound, and do the data support the conclusions?

Reviewer #1: Yes

Reviewer #2: Partly

2. Has the statistical analysis been performed appropriately and rigorously? 

Reviewer #1: I Don't Know

Reviewer #2: Yes

3. Have the authors made all data underlying the findings in their manuscript fully available?

Reviewer #1: Yes

Reviewer #2: Yes

4. Is the manuscript presented in an intelligible fashion and written in standard English?

Reviewer #1: Yes

Reviewer #2: No

5. Review Comments to the Author

Reviewer #1: Quite well written, interesting study on the use of FA to detect and predict obesity. However, several comments for either validating and/or improving the results:

1. In line 132, the comment was "I’m confused..! These acronyms (NW and OB) were not defined earlier. I assume they mean (normal weight and obese). Just define them". Later they were defined in line 136. Define them in line 132 instead.

2. In line 182, Using two softwares (MRtrix3 and FSL) may create multiple issues with your data (especially image header files). Make sure to check the gradient table as well as the image orientations (Left-Right, Anterior-Posterior, Superior-Inferior). I have a concern that the results may got affected and are misleading. You can test it by using only one subject to perform all analysis and track defined voxels (e.g. as a mask with a letter "R" in the right side of the image) + check all inputs and outputs (especially gradient table) and ensure that they are literally the same. In order to fully understand my point, please check this paper https://doi.org/10.1016/j.media.2014.05.012.

3. In section 2.1.3, How did you manage the high signals around the edges of the brain that resulted from eddy current distortions? The bright voxels surrounding the FA images that results from eddy current distortions should be removed before the alignment.

Reviewer #2: The overall premise of the study is quite intriguing. The relationship between neural connectivity and overall metabolic health is an important field of study.

Unfortunately, the stated hypotheses do not address the overall goal of the study. The approach is a "shotgun" method of searching for white matter tracts that may be associated with obesity. Which white matter tracts would the authors hypothesize to be associated with obesity and why?

Additionally, and as acknowledged by the authors in the Limitations section, the study sample is constrained to include only healthy females between 19-26 years old. This sample disallows generalizability to males, anyone older than mid-20s, and anyone considered "not healthy." Along the same line, the authors state that the study sample is representative of the general population within those demographics, but many factors lead to obesity that may lead someone to be described as unhealthy. Thus, it is unclear what group these subjects are actually representative of.

The authors repeatedly point out that their findings suggest there is neuroplasticity in these white matter tracts. I don't believe any recent work has suggested that neuroplasticity is not rampant across the brain for many reasons. Why would the findings of specific white matter tracts having an association with obesity, as opposed to all white matter tracts being associated, be the case? The authors rightly acknowledge they have found association and not causation, but they do pose any testable hypotheses that may explain their findings.

Finally, the manuscript is in significant need of editing for grammar, spelling, typos, and tense. Every paragraph has at least one error. Some of the manuscript is written in the first-person plural while other parts are in passive tense.

6. PLOS authors have the option to publish the peer review history of their article (what does this mean? ). If published, this will include your full peer review and any attached files.

**Do you want your identity to be public for this peer review?** For information about this choice, including consent withdrawal, please see our Privacy Policy .

Reviewer #1: No

Reviewer #2: No

---

## [Author Response · Author response to Decision Letter 1]

13 Dec 2024

Dear Editor and reviewers. We thank you for the possibility of addressing your comments and recommendations. First our comments to the editor’s comments:

Required Changes for Acceptance

1. Clarification and Definition of Acronyms

o Required Change: Define acronyms (e.g., NW for Normal Weight, OB for Obese) upon first use in the text (line 132), not after (line 136). Clear terminology improves readability and consistency for the reader.

Authors' response: The acronyms referred to have been corrected and defined in their first appearance.

2. Consistency in Software Use and Image Orientation

o Required Change: Address potential inconsistencies caused by using two different software tools (MRtrix3 and FSL). Perform a validation check as outlined by Reviewer 2 by testing the analysis with one subject, verifying that the gradient table and image orientations (Left-Right, Anterior-Posterior, Superior-Inferior) are consistent across all data processing steps. This is critical to ensure reliability in results and should be reported in the manuscript.

Authors' response: Following the indication made by Reviewer 1, a paragraph was added to the methods section indicating that the creators of the database implemented a preprocessing which consisted of validating the orientation of the diffusion gradient table after having used two different software tools (MRtrix3 and FSL). In fact, the algorithm applied by the creators of the database was based on the same article published by the authors Jeurissen et al. that Reviewer 1 suggested us to check.

o Conflict Resolution: If following only one software is not feasible, explicitly state the reasons and provide detailed explanations on how consistency across outputs was maintained and validated.

Authors' response: The AOMIC database was created by the authors after using the MRtrix3 and FSL software, so in the present work it was not possible to avoid using more than one software for the creation of the images studied. However, as indicated, the database was previously preprocessed by validating the orientation of the diffusion gradient table after using the two software.

3. Hypothesis Specification

o Required Change: Clarify the hypotheses regarding which white matter tracts may be associated with obesity and justify these choices based on relevant literature. The current “shotgun” approach should be refined to a more targeted hypothesis that strengthens the theoretical basis of the study.

Authors' response: Two objectives have been clarified in the introduction. One of them explicitly indicated which WM tracts were studied because they were reported in the literature as being usually associated with obesity in women. On the other hand, it was also indicated that, in addition to the study of the mentioned tracts, the rest of the tracts were also studied to allow the search for possible new findings in tracts not usually reported as being associated with obesity.

o Conflict Resolution: Align the hypothesis with the stated goal of understanding the relationship between specific white matter tracts and obesity.

Authors' response: Throughout the work, two objectives were systematically followed, which defined the division of work into two parts, indicating their respective two hypotheses and discussing whether these were finally fulfilled.

4. Sample Generalizability

o Required Change: Explicitly discuss the limitations of the sample (i.e., young, healthy females aged 19-26) and address how these constraints affect generalizability. Include a more precise description of how this sample is representative (or not) and clarify that the findings are specific to this demographic subset.

Authors' response: In the discussion section, it has been explicitly stated that the results obtained do not allow for generalization due to the restricted demographic characteristics of the subjects studied (right-handed young adult women with a medium level of education). However, it was also indicated that these restrictions were intentional since the objective was to study only the specified subjects. In the limitations section this has been pointed out, and also it was mentioned that future work should include the study of women and men with different age ranges, in order to later be able to discuss a possible generalization of the results to be obtained. On the other hand, in the methods section and within the description of the database, the statement that the subjects included were representative of the general population was eliminated to avoid possible confusion. This representation was only true considering the complete database. After the selection of specific subjects (right-handed young adult women with a medium level of education), the generalization would no longer be fulfilled. Also, this non-generalization was later pointed out in the discussion section.

Conflict Resolution: Ensure that any generalization to broader populations is cautious and adequately qualified to avoid overstatement.

Authors' response: In the discussion section, it has been explained that the results obtained cannot be generalized due to the characteristics of the subjects studied. In addition, statements in the initial sections that could cause confusion were deleted.

5. Neuroplasticity Discussion

o Required Change: Avoid broad statements about neuroplasticity being “rampant” and instead provide a more detailed analysis of why only certain white matter tracts might be associated with obesity. This analysis should align with the specific tracts examined and avoid implying unsupported causation.

Authors' response: Throughout the manuscript, most of the statements that alluded to a hypothesis of neuroplasticity were eliminated. This hypothesis has been reduced to two paragraphs in the discussion section, including new references to works related to the topic to support a suggested hypothesis on neuroplasticity, but clarifying that it can only be considered as speculation, in addition to pointing out the need for future work to have a broader discussion on the topic. It has also been indicated in the discussion that it is not possible to suggest a possible causal relationship based on the results reported in this work, reducing it only to correlation relationships with strictly statistical support.

o Conflict Resolution: Ensure statements about neuroplasticity are focused on observed associations within the study’s constraints and are not overly generalized.

Authors' response: In addition to reducing the number of statements about a neuroplasticity hypothesis in the manuscript, it has been indicated that both the reported results and the suggested hypotheses were related only to the subjects studied (young adult women with a medium level of education). Furthermore, in addition to supporting a hypothesis suggestion with the results reported in this work, new references to other works on neuroplasticity associated specifically with women were also added.

6. Language and Consistency

o Required Change: Address the manuscript’s language issues, including grammar, spelling, tense consistency, and voice. Ensure the manuscript is written in either first-person or passive voice consistently, not a mix of both.

Authors' response: A thorough review of the manuscript's language issues has been done, including grammar, spelling, tense consistency, and voice, in addition to writing the entire manuscript in the passive voice.

o Conflict Resolution: This is a crucial change for clarity and professionalism, as inconsistent language can undermine the study’s perceived rigor and readability.

Authors' response: The full manuscript was reviewed to ensure consistency in language.

Recommended Improvements (Not Required for Acceptance)

1. Data on Eddy Current Distortion Management

o Recommended Improvement: Detail how high signals from eddy current distortions were managed in the FA images (e.g., removing bright voxels at the edges of the brain). While not critical, explaining these steps will enhance the transparency and quality of data processing.

Authors' response: Within the preprocessing performed by the database creators, the eddy current correction was performed, including the high signals generated by these distortions. However, in order not to extend the description of the preprocessing performed by its creators, only a summary of the preprocessing steps was shown in the methods section, indicating that a more detailed description of them can be found on the database website and in their respective publication.

2. Refining the Neuroplasticity Interpretation

o Recommended Improvement: Provide a more nuanced discussion of neuroplasticity to address why only specific tracts may be associated with obesity, rather than assuming associations are widespread. This will improve the discussion’s scientific precision.

Authors' response: Two paragraphs in the discussion section were included, discussing the topic of neuroplasticity, citing various works that have reported neuroplasticity only in women, and suggesting a nuanced hypothesis, warning about its restrictions and explaining the future work that is necessary to give it greater support.

3. Spelling and Grammar in Depth

o Recommended Improvement: Conduct an additional layer of proofreading focused on minor language issues (e.g., typos, tense inconsistencies). This will improve readability but is supplementary to the broader language consistency required above.

Authors' response: A complete review has been done on possible language problems.

Specific Feedback on Manuscript Sections

1. Abstract and Introduction

o Ensure hypotheses are clearly stated and aligned with the study objectives. Define abbreviations when first used to enhance clarity.

Authors' response: Two main objectives were explicitly defined in the manuscript, indicating the work related to each one and their respective hypotheses.

2. Methods

o Provide a thorough description of all software used, including validation of consistency in data processing steps if using multiple software tools.

Authors' response: The description of the software used was extended. The necessary references were also provided to consult more details about the preprocessing performed by the authors of the database.

o Specify how high-signal voxels from eddy current distortions were handled.

Authors' response: Within the preprocessing done by the authors of the database, corrections were made on high intensities caused by eddy current distortions.

o Detail the justification for specific white matter tracts examined based on theoretical or empirical grounding to align with the study’s objectives.

Authors' response: Based on the literature, the tracts that have been commonly reported to be associated with obesity in women were specified in the introduction. It was indicated that, in this work, greater attention was paid to those tracts, and it was explained that other WM tracts not commonly reported were also studied in search of possible new findings.

3. Results

o Ensure all results reported strictly relate to the statistical analyses stated in the Methods section.

Authors' response: All results presented were supported by rigorous statistical analyses described in the methods section.

o Clarify the rationale for significant differences across white matter tracts and link these findings back to the hypotheses.

Authors' response: The significance of the statistical analyses was established by looking for statistical differences between tract measurements considering two study groups (normal and overweight/obese), by finding correlations between tract measurements and the BMI of the subjects, and by correlating the BMI of the subjects with the quantifications predicted by the created models. All the results supported the confirmation of the two hypotheses corresponding to the two objectives described in the work, respectively.

4. Discussion

o Ensure the neuroplasticity findings are discussed with nuance, avoiding generalizations across the brain and focusing instead on specific tracts related to obesity.

Authors' response: The neuroplasticity hypothesis was reported to be specific to some WM tracts found to be associated with obesity in women.

o Discuss the limitations of the study sample in terms of generalizability, clearly outlining how these limitations may affect broader applications of the findings.

Authors' response: The limitations of the work were mainly established by the restricted characteristics of the subjects studied (young adult women), implying future work studying both women and men, with different age ranges, in addition to conducting longitudinal studies to discuss possible processes of brain adaptation. Therefore, no generalizations were made, but the reported findings were intended to be the beginning of further work that, in the end, will allow the development of possible clinical applications.

o Ensure that interpretation statements, especially on neuroplasticity, are well-supported by the study’s data.

Authors' response: All interpretations of the results were supported by the findings obtained in the present work and by those reported by others. However, with respect to neoplasticity, statements on this topic were limited throughout the manuscript, warning that what is presented is a hypothesis, pointing out its limitations and restrictions, and indicating possible future work that is required for further discussion.

5. Conclusion

o Avoid overstating the findings’ generalizability to broader populations or implying causation where only associations were observed.

Authors' response: Due to the restricted characteristics of the subjects studied, no generalization statements were made in the conclusion section, and only correlational relationships and no causal relationships were reported.

Reviewer #1: Quite well written, interesting study on the use of FA to detect and predict obesity. However, several comments for either validating and/or improving the results:

Authors' response: We appreciate your positive feedback. We have considered all of your comments and tried to make the required corrections.

1. In line 132, the comment was "I’m confused..! These acronyms (NW and OB) were not defined earlier. I assume they mean (normal weight and obese). Just define them". Later they were defined in line 136. Define them in line 132 instead.

Authors' response: The acronyms referred to have been corrected and defined in their first appearance.

2. In line 182, Using two softwares (MRtrix3 and FSL) may create multiple issues with your data (especially image header files). Make sure to check the gradient table as well as the image orientations (Left-Right, Anterior-Posterior, Superior-Inferior). I have a concern that the results may got affected and are misleading. You can test it by using only one subject to perform all analysis and track defined voxels (e.g. as a mask with a letter "R" in the right side of the image) + check all inputs and outputs (especially gradient table) and ensure that they are literally the same. In order to fully understand my point, please check this paper https://doi.org/10.1016/j.media.2014.05.012.

Authors' response: A paragraph was added to the methods section indicating that the creators of the database implemented a preprocessing which consisted of validating the orientation of the diffusion gradient table after having used two different software tools (MRtrix3 and FSL). In fact, the algorithm applied by the creators of the database was based on the same article suggested by you and published by the authors Jeurissen et al.

3. In section 2.1.3, How did you manage the high signals around the edges of the brain that resulted from eddy current distortions? The bright voxels surrounding the FA images that results from eddy current distortions should be removed before the alignment.

Authors' response: Within the preprocessing performed by the database creators, the eddy current correction was performed, including the high signals generated by these distortions. However, in order not to extend the description of the preprocessing performed by its creators, only a summary of the preprocessing steps was shown in the methods section, indicating that a more detailed description of them can

---

## [Decision Letter · Decision Letter 1]

11 Feb 2025

Novel brain biomarkers of obesity in young adult women based on statistical measurements of white matter tracts

PONE-D-24-39635R1

Dear Dr. de Celis-Alonso,

We’re pleased to inform you that your manuscript has been judged scientifically suitable for publication and will be formally accepted for publication once it meets all outstanding technical requirements.

Kind regards,

Maryam Bemanalizadeh

Academic Editor

PLOS ONE

Additional Editor Comments (optional):

Reviewers' comments:

Reviewer's Responses to Questions

**Comments to the Author**

1. If the authors have adequately addressed your comments raised in a previous round of review and you feel that this manuscript is now acceptable for publication, you may indicate that here to bypass the “Comments to the Author” section, enter your conflict of interest statement in the “Confidential to Editor” section, and submit your "Accept" recommendation.

Reviewer #1: All comments have been addressed

2. Is the manuscript technically sound, and do the data support the conclusions?

Reviewer #1: Yes

3. Has the statistical analysis been performed appropriately and rigorously? 

Reviewer #1: I Don't Know

4. Have the authors made all data underlying the findings in their manuscript fully available?

Reviewer #1: Yes

5. Is the manuscript presented in an intelligible fashion and written in standard English?

Reviewer #1: Yes

6. Review Comments to the Author

Reviewer #1: (No Response)

7. PLOS authors have the option to publish the peer review history of their article (what does this mean? ). If published, this will include your full peer review and any attached files.

**Do you want your identity to be public for this peer review?** For information about this choice, including consent withdrawal, please see our Privacy Policy .

Reviewer #1: No

---

## [Editor Report · Acceptance letter]

PONE-D-24-39635R1

PLOS ONE

Dear Dr. de Celis-Alonso,

I'm pleased to inform you that your manuscript has been deemed suitable for publication in PLOS ONE. Congratulations! Your manuscript is now being handed over to our production team.

Kind regards,

on behalf of

Dr. Maryam Bemanalizadeh

Academic Editor

PLOS ONE